# Single-Line Multi-Channel Flexible Stress Sensor Arrays

**DOI:** 10.3390/mi14081554

**Published:** 2023-08-03

**Authors:** Jiayi Yang, Yuanyuan Chen, Shuoyan Liu, Chang Liu, Tian Ma, Zhenmin Luo, Gang Ge

**Affiliations:** 1College of Computer Science and Technology, Xi’an University of Science and Technology, Xi’an 710054, China; 2College of Safety Science and Engineering, Xi’an University of Science and Technology, Xi’an 710054, China; 3Department of Materials Science and Engineering, National University of Singapore, Singapore 117583, Singapore

**Keywords:** pressure sensor array, tactile sensor array, resonators, array integration, soft sensor array, force sensor array, parallel signal processing

## Abstract

Flexible stress sensor arrays, comprising multiple flexible stress sensor units, enable accurate quantification and analysis of spatial stress distribution. Nevertheless, the current implementation of flexible stress sensor arrays faces the challenge of excessive signal wires, resulting in reduced deformability, stability, reliability, and increased costs. The primary obstacle lies in the electric amplitude modulation nature of the sensor unit’s signal (e.g., resistance and capacitance), allowing only one signal per wire. To overcome this challenge, the single-line multi-channel signal (SLMC) measurement has been developed, enabling simultaneous detection of multiple sensor signals through one or two signal wires, which effectively reduces the number of signal wires, thereby enhancing stability, deformability, and reliability. This review offers a general knowledge of SLMC measurement beginning with flexible stress sensors and their piezoresistive, capacitive, piezoelectric, and triboelectric sensing mechanisms. A further discussion is given on different arraying methods and their corresponding advantages and disadvantages. Finally, this review categorizes existing SLMC measurement methods into RLC series resonant sensing, transmission line sensing, ionic conductor sensing, triboelectric sensing, piezoresistive sensing, and distributed fiber optic sensing based on their mechanisms, describes the mechanisms and characteristics of each method and summarizes the research status of SLMC measurement.

## 1. Introduction

Stress refers to the internal resistance or response that occurs within an object when it is subjected to external forces or loads. A stress sensor is an electronic device used to measure and detect mechanical strain or deformation in materials. These sensors convert the physical stress into measurable electrical signals [1,2,3]. Stress sensors find wide applications in engineering and materials science, where they are used to assess the integrity of structures, monitor deformations, and determine load distributions. Typically, these sensors utilize rigid materials such as metals and semiconductors. However, their limited ability to withstand significant deformations restricts their usefulness. With the advent of wearable electronic devices, flexible stress sensors have become more desirable due to their high sensitivity, good repeatability, and large stretchable range, making them ideal for wearable devices, human-computer interaction, intelligent robots, and health monitoring.

The research work of Professor Zhenan Bao and Professor John A. Rogers is of great significance in the field of flexible electronics, particularly in the development of high-performance flexible stress sensors [4,5,6,7,8,9,10,11,12]. Their research achievements have provided crucial technical support and guidance for stress sensors in flexible electronic applications. For instance, in their review, professor Rogers et al. [13]. introduced unconventional methods for fabricating and patterning nanomaterials, which laid a new scientific foundation for the fabrication of small-sized and patterned structures. These methods have significant implications in specific domains and in combination with other fabrication techniques. On the other hand, professor Bao’s research [14] has made significant progress in the scalable synthesis of multifunctional polyaniline hydrogels and their outstanding electrode performance, offering strong support for the preparation of flexible stress sensors. These studies provide important references for our understanding and application of flexible stress sensors.

Flexible stress sensor array, consisting of multiple flexible stress sensor units, can better quantify and analyze the magnitude and distribution of spatial stress, with advantages in flexibility, integration, and systematicity [15,16]. These sensors can be fabricated by conductive materials such as metal [17,18,19], carbon based [20,21,22], conducting polymer [23,24,25]. However, current implementations of flexible stress sensor arrays encounter the challenge of excessive signal wires, leading to reduced deformability, stability, and reliability, as well as increased costs [16,26,27]. While wireless signal transmission is commonly employed to reduce the number of signal wires between the sensor array and readout circuit, it does not address the issue within the flexible stress sensor array itself. The primary hindrance posed by the abundance of signal wires is the nature of the sensor unit’s electric amplitude modulation signal (e.g., resistance and capacitance), which restricts the transmission of only one signal per wire.

To address this challenge, the single-line multi-channel (SLMC) signal measurement has been developed, enabling simultaneous detection of multiple sensor signals through one or two signal wires [15,16,28,29,30,31,32,33,34,35,36,37,38,39,40]. By utilizing SLMC technology, the flexible stress sensor array can measure the magnitude and location of the stress by one or two signal lines. This approach effectively reduces the number of signal wires, enhancing stability, deformability, and reliability, and facilitating the application of flexible stress sensor arrays in bionic robots and rehabilitation medicine. Existing research has successfully implemented various mechanisms including RLC series resonant sensing, transmission line sensing, ionic conductor sensing, triboelectric sensing, piezoresistive sensing, and distributed fiber optic sensing [15,16,28,31,32,33,34,36,41,42,43,44,45,46]. These advancements have demonstrated promising potential in the domains of bionic robots, medical rehabilitation, and other related fields, as depicted in Figure 1.

## 2. Traditional Flexible Stress Sensor Array

Flexible stress sensors have significant potential in the fields of bionic robotics and rehabilitation medicine due to their outstanding flexibility and stretchability. Novel materials and structures are the most common approaches to enhance the performance of flexible stress sensors, which are determined by the mechanism [49]. Piezoresistive [50,51,52], capacitive [53,54,55], triboelectric [56,57], and piezoelectric [58] are the four mechanisms for flexible stress sensors, as shown in Figure 2.

### 2.1. Piezoresistive Flexible Stress Sensor Array Method

Piezoresistive flexible stress sensor consists of a piezoresistive material that converts mechanical stress into resistance when subjected to stress, resulting in a change in resistance by modifying the shape or relative position of the conductive filler, as shown in Figure 2a [50,59]. The equation for resistance can be expressed as:(1)$$R=\frac{\rho L}{wt}$$
where *R* is resistance, ρ is resistivity, *L*, ω and *t* are the length, width, and thickness of the resistor, respectively.

Several research works have been conducted to develop stress sensors using different materials and structures. Park et al. [60] developed a hyperelastic stress sensor, which is based on silicone rubber and eutectic gallium-indium (EGaIn). The sensor contains EGaIn in microchannels of the silicone rubber elastic body. The stress applied to the sensor causes deformation in the cross-sectional area of the EGaIn, which changes the resistance and enhances the sensitivity of the sensor. Figure 2a(i) is another example of EGaIn-based hyperelastic stress sensor [61]. As shown in Figure 2a(ii), Liao et al. [41] designed a sensor array that can measure stress position using conductive graphite film, paper, and an intermediate spacer. The sensor’s output resistance depends on the stress position from the connecting electrode, providing high sensitivity and accurate measurement of stress position. As shown in Figure 2a(iii), Shi et al. [62] developed a transparent and flexible stress sensor that utilizes polydimethylsiloxane (PDMS) and urchin-like hollow carbon spheres (UHCS). Stress-induced changes in the relative position of UHCS in PDMS lead to changes in resistance with ultrahigh sensitivity and high transparency. As shown in Figure 2a(iv), Yun et al. [63] created a mixed material stress sensor that contains PDMS, EGaIn, and iron powder. The sensor’s conductive fillers change their shape and position in response to stress, leading to changes in resistance. The conductive fillers return to their initial state when the stress is removed, and the initial resistance is restored. Figure 2a illustrates the different stress sensors developed in the works mentioned above.

**Figure 2 micromachines-14-01554-f002:**
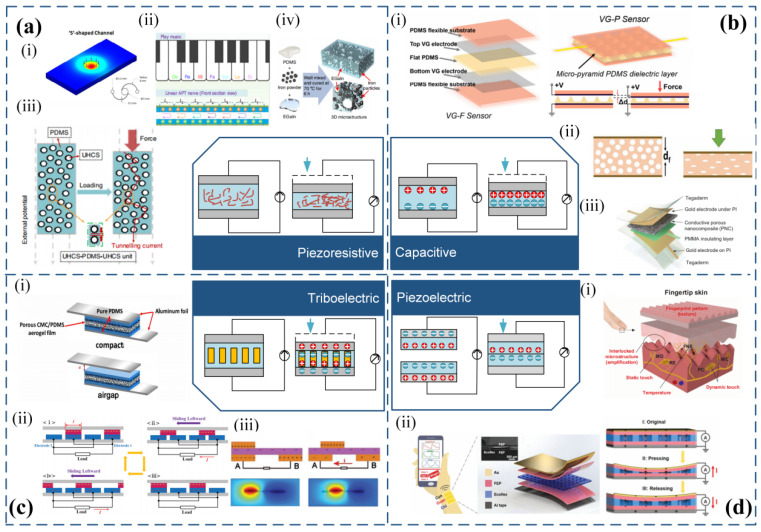
Sensing mechanisms and basic structures of flexible strain sensors. (**a**) Piezoresistive sensing mechanisms. (**i**) Cross-sectional area change in conductive filler eutectic gallium indium (reprinted with permission from [61]; copyright 2016, American Chemical Society). (**ii**) Change in relative position of biomimetic sensory transmission nerve (reprinted with permission from [41]; copyright 2020, Springer Nature). (**iii**) Change in relative position of polydimethylsiloxane (PDMS) and sea urchin-shaped hollow carbon spheres (reprinted with permission from [64]; copyright 2020, Springer Nature). (**iv**) Change in shape of conductive filler (PDMS, eutectic gallium indium, Fe) (reprinted with permission from [41]; copyright 2019, Springer Nature). (**b**) Capacitive sensing mechanisms. (**i**) Variable distance type capacitive sensor (reprinted with permission from [64]; copyright 2023, MDPI). (**ii**) Variable dielectric type capacitive sensor (reprinted with from [64]; copyright 2020, Wiley-VCH). (**iii**) Hybrid capacitive sensor combining variable distance and variable dielectric types (reprinted with permission from [64]; copyright 2021, Wiley-VCH). (**c**) Triboelectric sensing mechanisms. (**i**) Contact-separation type (reprinted with permission from [64]; copyright 2017, Elsevier). (**ii**) Contact-sliding type (reprinted with permission from [64]; copyright 2014, Wiley-VCH). (**iii**) Independent type (reprinted with permission from [65]; copyright 2018, Wiley-VCH). (**d**) Piezoelectric sensing mechanisms. (**i**) Multifunctional electronic skin based on interlocking structure (reprinted with permission from [34]; copyright 2015, AAAS). (**ii**) Piezoelectric composite based on stacking structure (reprinted with permission from [64]; copyright 2020, Wiley-VCH).

There are three methods available for implementing piezoresistive flexible stress sensor arrays. These include unit-independent measurement [66], row-column multiplexing measurement [67], and anisotropic electrical impedance tomography [68,69], which are illustrated in Figure 3.

The unit-independent measurement involves using independent electrodes and signal wires for each sensing unit. While this method has high measurement accuracy, it can reduce reliability due to the large number of signal wires required. As shown in Figure 3a, Wu et al. [66] utilized this method to design a sensor array for real-time monitoring of pressure and temperature in smart insoles, using a pressure-sensitive layer based on a mixed carbon nanotube/polydimethylsiloxane (CNT/PDMS) material and a pair of independent electrodes in each sensing unit. The row–column multiplexing measurement method arranges upper and lower electrodes in a cross pattern to reduce the number of signal wires. However, this method may be susceptible to signal crosstalk between sensor units. As shown in Figure 3b, Yu et al. [67] designed a stretchable and flexible stress sensor array for electronic skin, consisting of 6 × 6 sensing units, using a three-layer structure in each sensing unit: top electrode, stress-sensitive medium layer, and bottom electrode. The top and bottom electrodes adopt a cross-arrangement structure. The anisotropic electrical impedance tomography calculates the multi-dimensional resistivity distribution inside a composite material to measure the stress magnitude and position, eliminating the need for a complicated flexible electrode array on the sensor surface. However, this method may also be susceptible to signal crosstalk between sensor units, and the calculation of composite material resistivity can be complex. As shown in Figure 3c, Lee et al. [68] designed a stretchable 3D stress sensor array for human–machine interfaces that used this method. Applying stress increases the resistivity, enabling measurement of stress magnitude and position. As shown in Figure 3d, Duan et al. [69] proposed using anisotropic electrical impedance tomography on piezoresistive conductive fabric to realize low-cost and large-area touch sensing for dynamic touch stress mapping of single or multiple points.

**Figure 3 micromachines-14-01554-f003:**
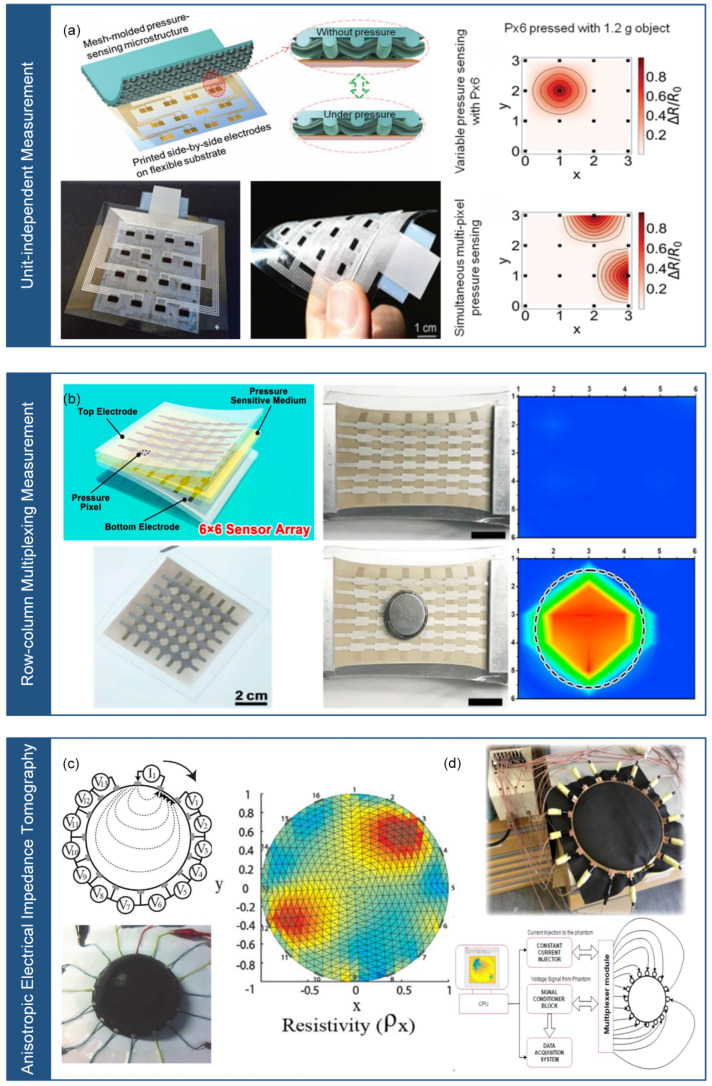
Array measurement methods for piezoresistive flexible stress sensors. (**a**) Unit-independent measurement (reprinted with permission from [70]; copyright 2020, Wiley-VCH). (**b**) Row-column multiplexed measurement (reprinted with permission from [70]; copyright 2020, Elsevier). (**c**) Anisotropic electrical impedance tomography (reprinted with permission from [70]; copyright 2017, Springer Nature). (**d**) Anisotropic electrical impedance tomography (reprinted with permission from [70]; copyright 2019, Springer Nature). Note: In (**c**,**d**), the same Anisotropic electrical impedance tomography is presented but from different references.

Piezoresistive flexible stress sensors are highly sensitive, offer a wide measurement range, are cost-effective, and have simple signal acquisition circuitry, making them a popular choice. These sensors can be integrated into arrays using techniques, including unit-independent measurement [66], row-column multiplexing measurement [67], and anisotropic electrical impedance tomography [68,69]. Despite their usefulness, these techniques have certain limitations, such as the requirement of numerous signal wires, intricate structures, and complicated acquisition circuits.

### 2.2. Capacitive Flexible Stress Sensor Array Method

The capacitive flexible stress sensor is composed of upper and lower electrodes separated by a compressible dielectric material [53,54,71]. Applying stress to the electrodes compresses the dielectric material, decreasing the distance between the two electrodes and converting stress into capacitance, as shown in Figure 2b [72]. The equation for the capacitive sensor can be written as:(2)$$C=\frac{\varepsilon_{r}\varepsilon_{0}A}{d}$$
where *C* represents the capacitance, $\varepsilon_{r}$ is the effective dielectric constant of the dielectric material, $\varepsilon_{0}$ is the vacuum dielectric constant, *A* is the overlap area of the electrodes, and *d* is the distance between the electrodes.

Applying stress changes the electrode overlap area (A) and spacing (d) of the capacitive sensor, resulting in a variation in capacitance [73,74]. Capacitive stress sensors are divided into two types: variable distance and variable dielectric sensors. The former involves one stationary electrode while the other moves relative to it. As shown in Figure 2b(i), Zhao et al. [75] proposed a micro-pyramid dielectric layer with graphene electrodes and PDMS as the dielectric layer, which increases sensitivity. In the variable dielectric type, a compressible dielectric material is placed on top of parallel finger-shaped electrodes, and stress changes the dielectric constant above the electrodes, resulting in a change in capacitance. As shown in Figure 2b(ii), Mahmoudinezhad et al. [76] designed a compressible stress sensor based on PDMS foam and finger-shaped electrodes. Our team proposed a flexible capacitive stress sensor based on liquid metal elastomer foam, which combines the variable distance and variable dielectric types. Liquid metal elastomer foam has a high dielectric constant that gradually increases during compression, providing high sensitivity and a wide measurement range. Recently, the Lu group proposed a hybrid response stress sensor that combines resistance and capacitance [77]. As shown in Figure 2b(iii), The sensor comprises porous nanocomposites (PNC) made of carbon nanotubes (CNT) and Ecoflex, placed between two parallel electrodes, and an ultra-thin insulating layer (500 nm) added between the PNC and one electrode. Applying stress transforms the PNC from a dielectric material to a conductive material, increasing the propagation distance of AC current within the PNC, and greatly reducing the effective distance between the two electrodes. This achieves high sensitivity and a large range for the capacitive stress sensor.

The measurement techniques for capacitive flexible stress sensor arrays are akin to those used for piezoresistive sensor arrays, and comprise unit-independent and row–column multiplexing measurements, as illustrated in Figure 4.

The unit-independent measurement involves independent top and bottom electrodes and signal wires for each sensor unit, providing high accuracy but reduced reliability due to the large number of signal wires. As shown in Figure 4a, Bae et al. [20] proposed a flexible sensor array with 4 × 4 units for real-time monitoring and discrimination of stress and temperature in electronic skin, where each sensing unit consists of a top part (Parylene C substrate, Ni/Ti electrodes, reduced graphene oxide thermistor, and dielectric layer) and a bottom part (Parylene C substrate and a microstructural CNT/PDMS). Stress changes the shape of the micro-protruding electrodes, reducing the electrode spacing and changing the capacitance. The proposed sensor array individually connects each sensing unit to two signal wires. In contrast, the row–column multiplexing measurement involves the cross-arrangement of the top and bottom electrodes, resulting in fewer signal wires but with the disadvantage of parasitic capacitance interference between sensing units [78]. Wang et al. [79] developed a flexible capacitive tactile sensor array for real-time contact force measurement in prosthetics, which integrates 8 × 8 sensing units, each consisting of four layers: copper electrodes arranged in a cross pattern, a PDMS film, and a PDMS bump layer for stress concentration. The sensor array is integrated into the prosthetic device, achieving real-time visualization of grip force. As shown in Figure 4b, Zhou et al. also proposed a capacitive stress sensor array based on the row–column multiplexing measurement [80].

**Figure 4 micromachines-14-01554-f004:**
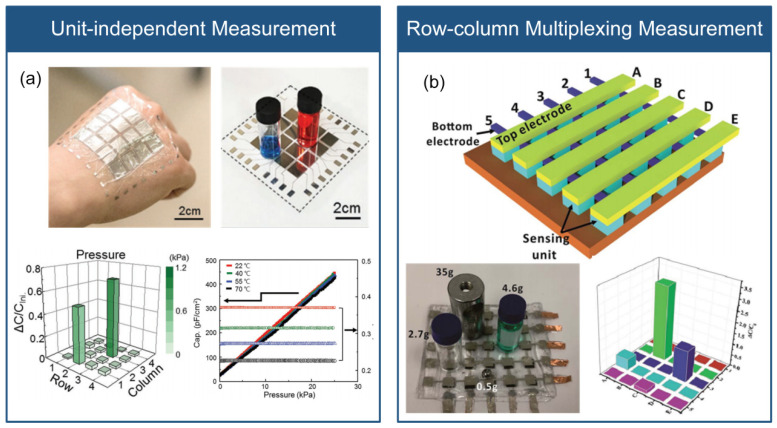
Array measurement methods of capacitive flexible stress sensors. (**a**) Unit-independent measurement (reprinted with permission from [70]; copyright 2018, Wiley-VCH). (**b**) Row–column multiplexed measurement (reprinted with permission from [80]; copyright 2019, The Royal Society of Chemistry).

Capacitive flexible stress sensors are advantageous due to their good temperature independence, low power consumption, high sensitivity, compact circuit layout, and simple device structure. Flexible capacitive stress sensor arrays can be fabricated using unit-independent measurement [20] and row–column multiplexing measurement [79]. Nevertheless, similar to the flexible piezoresistive stress sensor array, these methods have drawbacks such as crosstalk, multiple signal wires, and a complicated structure.

### 2.3. Triboelectric Flexible Stress Sensor Array Method

The triboelectric flexible stress sensor utilizes the coupling effect of frictional electrification and electrostatic induction to generate and convert mechanical energy into electrical energy for stress measurement [56]. As shown in Figure 2c, the sensor comprises two materials with different electron affinities. When the two different friction materials make contact, their varying electron affinity leads to a difference in electron attraction ability, resulting in the formation of a potential difference. The material with stronger electron affinity has a negative surface charge, while the material with weaker electron affinity has a positive surface charge. Electrons flow through an external circuit to generate current. The triboelectric nanogenerator (TENG) comprises three structures: contact-separation type, contact-sliding type, and independent type. In the contact-separation type, two materials with different electron affinity face each other, and external force causes them to contact perpendicularly, generating a potential difference that forms a current after the external force is removed. Figure 2c(i) shows a TENG based on a high-polymer aerogel film proposed by Tang et al. [81] to improve the performance of the contact-separation type TENG. The porous structure of the aerogel produces a larger potential difference at the contact surface, increasing the output of the electric signal. The contact-sliding type generates current by repeatedly sliding two seamlessly contacting materials laterally relative to each other, polarizing the electrode. Figure 2c(ii) depicts a TENG with a linear grating structure proposed by Xie et al. [82], which achieved up to 85% total conversion efficiency at low operating frequencies. The independent type differs slightly from the contact-sliding type. It comprises two electrodes placed on the same horizontal plane and separated by a certain distance. Another piece of insulating film acts as a slider that seamlessly contacts the two electrodes, generating charge on the slider surface. Zhu et al. [83] designed an independent TENG with a rotating structure, which fixes the independent friction layer above the two electrodes and rotates to produce a periodic potential difference, allowing electrons to flow between the two fixed electrodes. The sensor has an output power of up to 1.5 W, and an energy conversion efficiency of 24%. Figure 2c(iii) shows another example of a TENG with a rotating structure [65].

The triboelectric flexible stress sensor array measurement includes two approaches: the unit-independent measurement [84] and the row-column multiplexing measurement [70], as shown in Figure 5. The unit-independent measurement refers to each sensor unit having an independent upper and lower triboelectric material and signal wire. Its advantage lies in high accuracy, while the disadvantage is the poor reliability caused by the large number of signal wires. As shown in Figure 5a, Zhou et al. [84] prepared a triboelectric nanogenerator with high sensitivity and high stretchability based on thermoplastic polyurethane (TPU) and silver nanowire/reduced graphene oxide (AgNWs/rGO). The authors proposed a 5 × 5 sensor array, and each sensing unit is connected by a separate AgNWs/rGO conductive layer. Stress makes the independent polymer materials in contact, producing an output voltage signal that indicates the magnitude and position of the stress. The row–column multiplexing measurement refers to the upper and lower layers of triboelectric materials arranged in a cross pattern, which reduces the number of signal wires, but introduces signal interference between sensing units. As shown in Figure 5b, Ning et al. [70] prepared a fiber-shaped triboelectric nanogenerator with ultra-high flexibility and stretchability based on silver nanowire/carbon nanotube and PDMS. The authors proposed an 8 × 8 sensor array, and each sensing unit includes cross-arranged fiber-shaped friction layers. When touching the sensor array, the corresponding pixels of the tactile sensor array produce output voltage signals that can reflect the tactile trajectory and stress distribution, showing promising applications in human–machine interaction and health monitoring.

The triboelectric flexible stress sensor array method can convert mechanical energy into electrical energy, which could potentially solve the issue of frequent recharging or battery replacement for wearable electronic devices. They offer advantages such as energy efficiency, good dynamic performance, low cost, a wide range of material options, and broad applications. However, triboelectric flexible stress sensor requires continuous friction to generate charges and are unable to detect static stress. Triboelectric flexible stress sensor arrays can be achieved through unit-independent measurements [84] and row–column multiplexing measurements [70]. Similar to capacitive sensors, these arraying methods have disadvantages such as multiple signal wires, complicated structures, and complicated acquisition circuits.

### 2.4. Piezoelectric Flexible Stress Sensor Array Method

Piezoelectric flexible stress sensors are made of materials that exhibit the piezoelectric effect. When stress is applied to these materials, they polarize, accumulate, or release charges, and convert stress into charge, enabling stress measurement [58]. Figure 2d shows commonly used piezoelectric materials, including single crystals, ceramics, and polymers. In addition, flexible materials such as porous polypropylene (PP) and porous polyvinylidene fluoride (PVDF) are also utilized. Park et al. [70] developed a multifunctional flexible electronic skin, shown in Figure 2d(i), based on PVDF and RGO with an interlocking microstructure and fingerprint-like patterns. This electronic skin can detect both static and dynamic tactile signals. Chu et al. [64], as shown in Figure 2d(ii), utilized piezoelectric polymers to create piezoelectric composites with a superposition structure, allowing for real-time monitoring of pulse signals at the radial artery and promising applications in wearable human health monitoring.

Piezoelectric flexible stress sensor arrays can be measured using two methods: row–column multiplexing measurement and shared electrode measurement, as illustrated in Figure 6. The row-column multiplexing measurement involves arranging the upper and lower electrodes in a cross pattern, which reduces the number of signal wires but leads to signal crosstalk between sensing units. To solve this issue, as shown in Figure 6a, Lin et al. [85] developed a flexible piezoelectric tactile sensor array with 25 sensing units arranged in a 5 × 5 matrix. They employed a double-layer comb-shaped structure for the row and column electrodes to eliminate signal crosstalk and reduce the number of connecting lines. Each sensing unit was composed of two protective layers (PDMS), two sensing layers (PVDF), and one insulation layer (PDMS). By applying stress to the protective layer, the sensing layer could convert the stress into charge, facilitating the identification of stress and position. This sensor array exhibited zero crosstalk, fast response time, high sensitivity, and durability. As shown in Figure 6b, Liu et al. [86] also developed a piezoelectric tactile sensor array, which used PVDF fibers and comprised two orthogonal arrays of 5 × 5 based on nanofiber-reinforced polyurethane films. This sensor array enabled real-time localization of the pressing position and tracking of the pressing trajectory, offering high resolution and flexibility. On the other hand, the shared electrode measurement involves independent electrodes and signal wires for each sensing unit, while sharing the same electrode and signal wire. This method provides high accuracy but still requires many signal wires. Liu et al. [87] presented a piezoelectric tactile sensor array based on PDMS and aluminum. The array consisted of four sensing units arranged in a 2 × 2 matrix, and each sensing unit was composed of a PDMS protrusion layer, a PET strip layer, a shared aluminum (Al) electrode layer, a PVDF film layer, an Al electrode layer, and a PET strip layer. The upper electrode layer, PVDF film layer, and lower electrode layer formed the piezoelectric sensing unit, with the four units sharing the same upper electrode. By applying three-dimensional stress to the top of the protrusion, different charge changes were generated in the four sensing units, enabling the measurement of normal and tangential stress and identification of roughness at variable speeds. As shown in Figure 6c, Yu et al. present a piezoelectric sensor array based on shared electrode method [88].

Piezoelectric flexible stress sensors offer advantages such as high sensitivity, good dynamic performance, and a simple structure. However, they are susceptible to temperature changes, leading to poor temperature stability. Similarly to flexible triboelectric stress sensors, piezoelectric flexible stress sensors are unable to measure static stress. The piezoelectric stress sensors array also present challenges, including a high number of signal wires, complicated structures, and complicated acquisition circuits.

In summary, piezoresistive, capacitive, triboelectric, and piezoelectric flexible stress sensors have all been employed in sensor arrays. While these methods fulfill the requirements for spatial stress distribution and high spatial resolution, increasing the integration levels in existing methods results in a significant rise in the number of signal wires, which decreases in flexibility and stretchability of the sensor array, along with increased complexity and cost. To address these issues, researchers have explored a novel sensor array measurement method known as the single-line multi-channel (SLMC) measurement. This approach utilizes one or two signal wires to simultaneously measure signals from multiple sensors, thereby reducing the number of signal wires required for the sensor array. The SLMC measurement has the potential to enhance the deformability, reliability, and stability of flexible stress sensor arrays, facilitating their practical application.

## 3. Single-Line Multi-Channel Measurement

SLMC measurement can simultaneously measure multiple sensor signals using one or two signal wires. Depending on the mechanism, SLMC measurement can be classified into various types, including RLC resonant sensing [15,28,29,32], transmission line sensing [16], ionic conductor sensing [33,36], triboelectric sensing [42,43], piezoresistive sensing [41], and distributed fiber optic sensing [34].

### 3.1. RLC Resonant Sensing

The RLC resonant sensing utilizes the interaction between inductance and capacitance. By adjusting the resonant frequency of the RLC resonant circuit, it is possible to obtain multiple sensor signals using only two signal lines. Implementing this technology requires selecting appropriate inductance and capacitance components, as well as designing and fabricating the circuit. An RLC resonator consists of resistors, inductors, and capacitors connected in series, forming a resonant circuit with a specific frequency and frequency-selective capability. By parallel connecting multiple RLC resonant circuits and adjusting their resonant frequencies, a multi-frequency resonant signal can be generated. Each resonant frequency corresponds to the output characteristics of a sensor unit, which can modify the resonant frequency or amplitude of the multi-frequency signal by converting stress into changes in resistance, inductance, or capacitance, as shown in Figure 7.

As shown in Figure 7a, Wen et al. [15] developed a flexible wireless passive LC stress sensor using a flexible composite protein film as the substrate. By adding regenerated silk fibroin film, which makes better air permeability and water permeability. The sensor employs a commercial printed circuit board and magnetron sputtering methods to create the inductor and capacitor in the LC circuit. The study combines radio frequency identification technology and the LC resonant circuit to design a wireless insole sensor for measuring sole stress distribution. The RLC series resonant sensing simplifies sensor wiring and reduces costs. As shown in Figure 7b, inspired by the somatosensory system, Lee et al. [28] proposed a stress sensor system using a wireless system to differentiate parallel signals. The sensor system comprises multiple stress sensors with polypyrrole-coated microstructured PDMS placed on top of electrodes, which operates as a capacitive sensor. By altering the number of turns of the coil, the system assigns specific resonant frequencies to each sensor. A stacked coil structure is adopted to conserve space and minimize crosstalk between coils. The sensor can simultaneously receive resonant signals from multiple stress sensors and utilize a convolutional neural network to train data for simultaneous measurement of stress magnitude and location. As shown in Figure 7c, Kim et al. [32] introduced a flexible sensor array composed of multiple bandpass filters. The array consists of parallel bandpass filters (RLC resonant circuits) and two wires. Each sensing unit corresponds to a bandpass filter, which can be assigned different filtering frequencies by adjusting the values of inductance and capacitance. The sensor array consists of an inductor, a capacitor, and a variable resistor made of a microfluidic channel filled with EGaIn. This enables separate measurements while sharing the same signal wire. The article also demonstrates the application of the flexible sensor array in tactile sensing and foot pressure measurement.

**Figure 7 micromachines-14-01554-f007:**
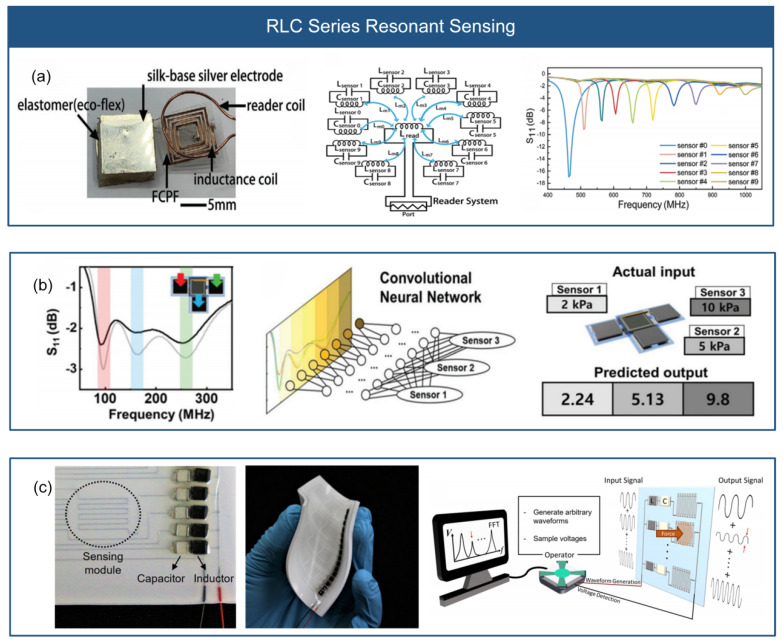
RLC series resonant sensing. (**a**) Wireless passive LC stress sensor (reprinted with permission from [15]; copyright 2021, Wiley-VCH). (**b**) Stress sensor system based on wireless system to differentiate parallel signals (reprinted with permission from [28]; copyright 2019, Wiley-VCH). (**c**) Flexible sensor array based on multiple band-pass filters (reprinted with permission from [32]; copyright 2022, Springer Nature).

The RLC resonant circuit allows for the selection of a specific frequency, facilitating the acquisition of multiple sensor signals using only two signal wires. By connecting RLC resonant circuits in parallel and series while adjusting their resonant frequencies, multiple resonant frequencies can be generated. Each resonant frequency in the multi-frequency resonant signal corresponds to the output characteristic of a sensor unit. The flexible sensor array based on RLC resonant sensing utilizes only two external wires to scan the sensor array, regardless of the number of sensors. The RLC sensing has the advantages of two signal lines, tunable sensitivity, and multi-model sensing. However, due to the RLC resonant circuit, the sensor array has a complicated circuit design, and has a crosstalk between different RLC resonators. RLC resonant sensing has wide applications in areas such as wireless communication, biomedical, and environmental monitoring. In wireless sensor networks, it enables wireless monitoring and data transmission of various physical parameters [31] (temperature, humidity, etc.). In bio-sensors, it can be used to monitor physiological parameters [5] and disease indicators within the human body.

### 3.2. Transmission Line Sensing

The transmission line sensing measures the voltage or current on the transmission line to obtain information about the target physical quantity. When the target physical quantity changes, it affects the electromagnetic field distribution on the transmission line, thereby altering the characteristics of the transmission line, such as resistance, inductance, and capacitance. Implementing this technology requires selecting appropriate transmission line types and parameters, as well as designing and manufacturing the sensors. The transmission line, composed of multi-core cables, coaxial cables, optical fibers, or wireless devices, can transmit electromagnetic energy. The transmission performance is influenced by several factors, such as transmission rate, signal bandwidth, data transmission quality, and anti-interference capability. As the working frequency increases and the wavelength decreases, the voltage and current on the transmission line vary with spatial position, resulting in wave-like fluctuations. These fluctuations of the electromagnetic signal on the transmission line can indicate the magnitude of stress at different positions, enabling the achievement of a SLMC measurement.

Leber et al. [16] proposed a stretchable transmission line for measuring stress at various positions using a single line. This transmission line is created by enclosing a liquid metal (65% Ga, 22% In, and 10% Sn) in a thermoplastic elastomer dielectric (pol(styrene-b-(ethylene-co-butylene)-b-styrene), SEBS) via thermal stretching and uses time-domain reflectometry to send high-frequency pulses. Stress magnitude and location are determined by analyzing the reflection resulting from applied stress. The stretchable transmission line exhibits high sensitivity, and repeatability, making it suitable for measuring stress distributions in electronic textiles, wearable electronic devices, and electronic skins. This approach offers several advantages, including complexity reduction and high productivity.

In transmission line sensing, the reflected electromagnetic pulse signal conveys information about the magnitude of stress at different locations by exploiting the fluctuations of voltage and current along the spatial position of the transmission line. The transmission line sensing only has one signal line, which has the advantage of high integration. While it has drawbacks, including high cost, complicated fabrication, and complexity of information decoding.

### 3.3. Ionic Conductor Sensing

The ionic conductor sensing utilizes the contact resistance and capacitance between the human hand and the ionic conductor under AC current. Ionic conductors can be made from ion-containing hydrogels or organic hydrogels. When exposed to high-frequency AC current, human touch on the ionic conductor causes changes in the contact resistance and capacitance within the electric double layer structure of the ionic conductor. The contact resistance is measured to determine the position of touch, while the newly formed contact capacitance after human touch is used to measure the magnitude of stress, as shown in Figure 8. The ionic conductor sensing enables the determination of stress magnitude and location information through two single-signal lines, providing a low-cost, easily operated for measuring stress distribution in electronic skins, wearable electronic devices. In addition, implementing this technology requires the selection of suitable ion conductor materials and the design of corresponding circuits.

Gao et al. [33] developed a pressure-sensitive touchpad with self-healing properties using an ionic conductor, as shown in Figure 8a. The ionic conductor is a polyzwitterion-clay nanocomposite hydrogel, which has flexibility, stretchability, transparency, and pressure- sensitivity. When touched, the hydrogel exhibits changes in the resistance and capacitance of the electric double layer structure, enabling the measurement of touch position and magnitude. The touch position is determined by the resistance, whereas the capacitance is dependent on the distance between the touch point and the electrode. This study provides an innovative solution for human–machine interfaces with self-healing properties of polymer nanocomposite hydrogels, along with the recognition and differentiation of multi-touch signals. As shown in Figure 8b, Wu et al. [36] proposed a multifunctional, wearable, and transparent human–machine interface based on ionic hydrogels. An anti-drying sodium alginate/polyacrylamide/ionic liquid double-network organohydrogel was synthesized using a wet spinning method. The interface utilizes a parallel mutual detection structure that allows for both single-point touch and multi-point touch using only two electrodes. The interface is composed of a pair of horizontal or spiral ionic hydrogel strips embedded in a flexible substrate. The ionic hydrogel exhibits excellent stretchability, transparency, and durability. When a finger touches the interface, the ionic hydrogel between the electrode and the touch point acts as a resistor that is connected to the circuit and is in series with the electric double-layer capacitor. This configuration allows for the determination of touch position and magnitude through variations in current and capacitance. This study employs only two electrodes for touch sensing, which simplifies maintenance and operation. Furthermore, the detection of multi-touch enhances the diversity of human–machine interaction and advances the development of the Internet of Things.

**Figure 8 micromachines-14-01554-f008:**
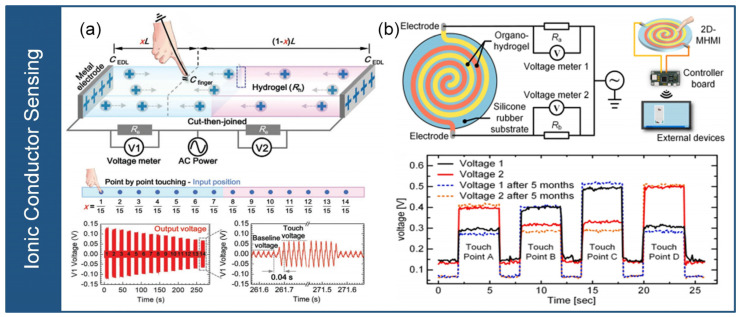
Ionic conductor sensing. (**a**) Self-healing composite hydrogel sensors (reprinted with permission from [33]; copyright 2020, Wiley-VCH). (**b**) Parallel interdigitated electrode structure-based hydrogel sensors (reprinted with permission from [36]; copyright 2021, Elsevier).

The ionic conductor exhibits varying resistances and capacitances based on the magnitude and location of the touch when it contacts with the human body. The resistance of the sensor indicates different positions on the sensor, whereas the capacitance corresponds to the intensity of the touch force. Two-dimensional planar touch sensing can be accomplished by utilizing four wires, offering the benefits of a streamlined circuit layout and a straightforward device structure. The ionic conductor sensing technology has the advantages of two signal lines and multi model measurement. However, it still needs to address challenges such as low temperature stability and complicated signal decoding. The ionic conductor sensing finds application potentials in fields such as human–machine interfaces and biosensing.

### 3.4. Triboelectric Sensing

Triboelectric sensing is a sensing array composed of triboelectric materials with unique physical and chemical properties (electron affinity, mobility, adhesion energy, and surface potential), which measures the change in the applied stress through the charge variation generated by the friction between the triboelectric materials. These materials are arranged in a way that generates triboelectric signals exhibiting diverse voltage or charge density. When subjected to triboelectric force, the different positions of the materials produce distinct signal responses, enabling the simultaneous detection of multiple frictional positions, as shown in Figure 9.

To reduce the number of connection wires and achieve high spatial resolution, Jang et al. [42] proposed a wearable single-ended triboelectric tactile sensor array that can detect multi-point friction, as shown in Figure 9a. This array consists of different stretchable block copolymer elastomeric dielectrics that are patterned with UV light. By adjusting the UV exposure time of the thermoplastic block copolymer film, triboelectric patterns with varying relative charge densities can be generated. With just one electrode, the tactile sensor array can discern the touch position. Guo et al. [43] developed a self-powered electronic skin based on an Archimedean spiral structure, as shown in Figure 9b. The electronic skin comprises a triboelectric layer (such as polymer, metal), patterned AgNW electrodes, and a PDMS substrate, capable of determining the frictional position without the need for an external power supply. These materials are distributed on the electrodes through a multilayer alignment transfer process. By designing the electrode with an Archimedean spiral structure, the triboelectric signal is distinguished into digital “0” or “1” based on the intrinsic electron affinity of the materials. Only four electrodes are needed to identify 280 frictional positions. The electronic skin easily integrable into portable electronic devices, such as laptops and healthcare devices, showing application potential in the fields of human–machine interface and artificial intelligence.

**Figure 9 micromachines-14-01554-f009:**
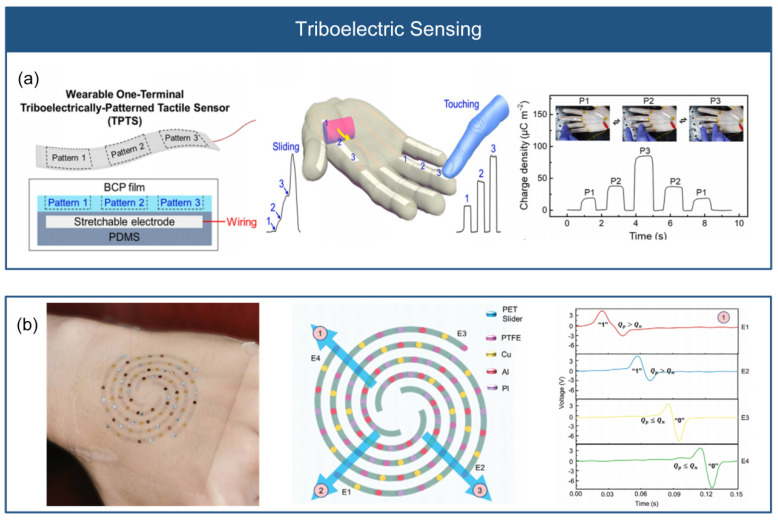
Triboelectric sensing. (**a**) Wearable single-ended frictional tactile sensor array (reprinted with permission from [42]; copyright 2022, Elsevier). (**b**) Self-powered electronic skin based on Archimedean spiral structure (reprinted with permission from [43]; copyright 2021, Wiley-VCH).

Applying friction to triboelectric sensors generates voltage or charge density based on the different friction positions and the triboelectric materials with different properties, therefore enabling SLMC measurement. Triboelectric sensing technology has the advantages of two signal lines, low energy consumption, high dynamic performance, but it still needs to solve the challenges of poor static measurement and low signal stability. Triboelectric sensing has the application potential in the fields of friction and wear monitoring and tactile sensing.

### 3.5. Piezoresistive Sensing

The piezoresistive sensing consists of materials with piezoresistivity that transforms the stress position into a corresponding resistance value. Applying stress at different positions leads to different resistance variation, which enables the realization of a SLMC measurement, as shown in Figure 10.

As shown in Figure 10a, Liao et al. [41] introduced a biomimetic sensory transmission neural sensor array capable of accurately determining the position of stress. The sensor array consists of a conductive graphite film, a grid paper, and a sheet in between. The proposed sensor array, known as the artificial perception and transmission neural sensor array, employs an independent two-layer electrode structure with progressively decreasing resistance in each electrode. Applying stress contacts the two electrodes, and the resulting resistance determines the stress position. The sensor array exhibits remarkable attributes such as rapid response, robustness, durability, and flexibility, enabling diverse applications such as multi-functional touch interaction. Furthermore, by integrating the spatiotemporal resolution function of the sensor array with artificial intelligence algorithms, this study demonstrated its applicability in neural prostheses and robots. The sensor array designed with a distinct double-layer structure, effectively resolves challenges related to complicated structures, intricate interconnections, and signal transmission interference.

**Figure 10 micromachines-14-01554-f010:**
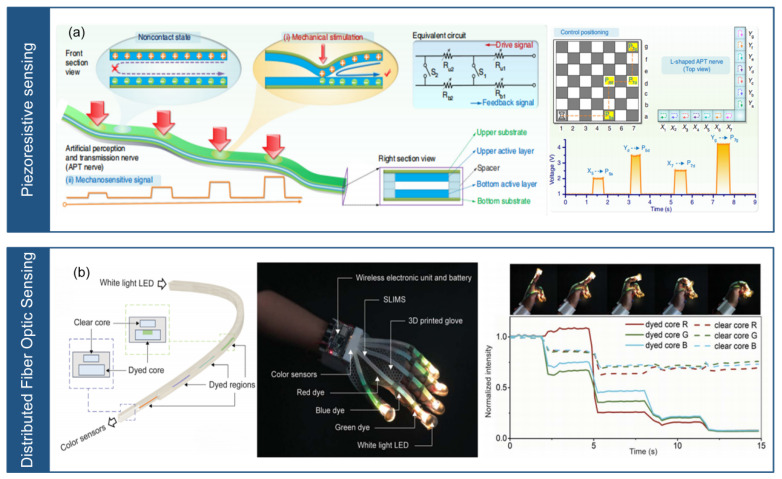
Piezoresistive sensing and distributed fiber optic sensing. (**a**) Biomimetic sensory nerve sensor (reprinted with permission from [41]; copyright 2020, Springer Nature). (**b**) Stretchable distributed fiber optic sensor (reprinted with permission from [34]; copyright 2020, AAAS).

By manipulating the resistance at different positions within the piezoresistive material, piezoresistive sensing-based sensor array can determine the stress location by analyzing the resulting resistance. Piezoresistive sensing has several advantages, such as one signal line, simple structure, low cost, and simple signal acquisition circuit. However, for some application scenarios, it is necessary to consider its stress magnitude and multiple-channel unmeasurably. Piezoresistive sensing technology is widely used in the field of structural monitoring. The sensors can be used to monitor stress changes in multiple key locations such as buildings or bridges to assess the health and safety of the structure. In the medical field, piezoresistive sensing technology can be applied to the measurement of stress distribution on human body, such as on mattresses, to improve treatment.

### 3.6. Distributed Fiber Optic Sensing

Distributed fiber optic sensing utilizes the continuous one-dimensional characteristic of optical fibers. The distributed fiber optic sensing relies on optical time-domain reflectometry, which is based on the principle of backscattering of light emitted by a source as it propagates through the fiber. The intensity of the backscattered light decreases in proportion to the distance and time traveled. By detecting changes in the time and intensity of the backscattered light, an optical detector can determine the magnitude of stress at different positions along the fiber, as shown in Figure 10.

As shown in Figure 10b, Bai et al. [34] developed a stretchable fiber optic sensor based on silicon dioxide. The sensor consists of a dual-core lightguide with one dyed core doped with absorbing dyes at discrete locations and one clear core, separated by silicone cladding. The sensor can measure the location, magnitude, and mode (stretching, bending, or twisting) of deformations using an elastic optical waveguide containing continuous or discrete colored patterns. Applying stress changes the chromaticity and intensity of the waveguide, allowing the position and magnitude of the stress to be determined through total internal reflection and absorption phenomena. The researchers successfully integrated this stretchable fiber optic sensor into a glove and demonstrated its effectiveness in monitoring various hand movements.

Applying stress to the distributed fiber optic sensor modifies the reflection and refraction of light sources within the fiber, resulting in variations in the detection time and intensity of the light source. These variations can indicate the magnitude and position of stress on the sensing array, making it possible to implement SLMC measurement. Distributed fiber optic sensing provides significant advantages, including one signal line, high spatial resolution, and multi-model detection. However, it also faces challenges such as fiber brittleness, signal parsing and data processing complexity.

Fiber Bragg Grating Sensors (FBGs) are used to realize fiber sensing by introducing a periodic index modulation structure into the fiber. Specifically, FBGs are composed of refractive index periodic modulation regions in optical fibers that cause specific wavelengths of light to be reflected or transmitted. Applying stress or temperature to FBGs changes their periodic refractive index structure, resulting in a shift in the reflected or transmitted wavelength, and then the stress or temperature is measured. FBGs has the advantages of high precision, fast response and anti-electromagnetic interference, and has been widely used in the fields of structure monitoring, pipeline inspection, aerospace and medical diagnosis. However, higher costs and flexibility limitations are some of the drawbacks of FBGs. Li et al. [89] proposed a sensor manufacturing method for joint motion monitoring in stroke rehabilitation by embedding silicon fiber grating in silicone tubes. The materials used include silicon fiber FBG and silicon tubes. The sensor has demonstrated excellent performance in detecting and differentiating joint movements, making it a promising medical tool for rehabilitation of stroke patients and potential detection of Parkinson’s tremor in stroke survivors.

Intensity modulation polymer optical fiber (POF) sensor is a fiber based sensing technology, its working principle is based on the intensity modulation of the optical signal. The sensor uses a sensitive material or structure that causes the intensity of the light signal to change when affected by the measured parameters. This intensity change can be measured and analyzed by the photodetector, so that the measurement parameters can be monitored and measured. Intensity modulated POF sensors have the advantages of simplicity, economy and high sensitivity, and are suitable for applications that measure changes in the intensity of optical signals, but are limited by changes in light intensity and the influence of ambient light [90].

FBGs uses the grating structure in fiber to measure optical signals, which is suitable for point measurement and has high spatial resolution and complicated reading system. The distributed optical fiber sensor is suitable for continuous measurement, with low spatial resolution and simplified reading principle. Intensity modulated POF sensors measure by adjusting the intensity of the optical signal, which is different from FBGs and distributed optical fiber sensors in working principle, spatial resolution and application field.

We summarized the advantages and disadvantages of the existing single-wire multi-channel measurement technology, as shown in Table 1.

In summary, the SLMC measurement reduces the need for multiple signal wires, enhancing the flexibility, stability, and reliability of flexible stress sensor arrays while also reducing costs. Additionally, the SLMC measurement introduces new concepts to the fundamental theory of measuring flexible stress sensor arrays, overcomes technical challenges, and promotes widespread adoption of soft stress sensor arrays.

## 4. Summary and Prospect

Flexible stress sensor arrays have gained popularity due to their high sensitivity, functionality, deformability, and stretchability. However, the sensor signal is an electric amplitude modulation signal (such as resistance and capacitance), which cannot simultaneously distinguish multiple signals through a single-signal wire. Thus, a large number of signal wires are needed to meet the measurement requirements over a large measurement area, which greatly increases costs, decreases equipment reliability and operability, and hinders their widespread application in wearable electronic devices such as electronic skin and health monitoring.

In this review, we introduce the stress sensing mechanisms based on piezoresistive, capacitive, triboelectric, and piezoelectric principles, as well as their array structures and performance. We focus on SLMC measurement that can transmit multiple signals through one or two signal wires. According to the mechanism of SLMC measurement, we classify the methods into RLC resonant sensing, transmission line sensing, ionic conductor sensing, triboelectric sensing, piezoresistive sensing, and distributed fiber optic sensing. We describe the mechanisms and characteristics of each method and summarize the research status of SLMC measurement.

Several techniques mentioned in this article have successfully verified the feasibility and reliability of the SLMC measurement, which reduces the number of signal wires, reduces costs, and improves the flexibility, stability, and reliability of flexible sensor arrays. Nevertheless, the scalability of each technology from sensors to large-scale commercial applications requires improvement. We can promote the application of SLMC signal transmission methods on flexible sensor arrays by optimizing structural design, material selection, and signal processing.

In previous studies, multiple sensing units were concatenated to achieve multimodal measurement, resulting in poor operability, reliability, and efficiency. To improve operability and efficiency, simple structural designs such as vertical stacking structures can be used to achieve higher-dimensional measurement [91,92,93]. Wearable electronic devices, such as electronic skin, are vulnerable to damage due to their flexible material, which negatively impacts usability and reduces efficiency. The use of self-healing polymer materials, such as composite hydrogels with self-repairing properties, can provide quick self-repair during use, which improves the stability and durability of wearable electronic devices [94,95,96]. Previous signal processing was based on simple threshold judgments, which is inadequate for signal decoupling that is affected by noise, signal damage, or multiple signals. The use of machine learning algorithms can improve the accuracy and efficiency of sensing. In conclusion, SLMC measurement is a new breakthrough in signal processing for flexible stress sensor arrays [27,97,98]. By optimizing structural design, material selection, and signal processing, they can promote implementation in health monitoring, human–machine interaction.

The SLMC measurement technologies reduce the number of signal lines, enhancing the deformability, stability, and reliability of the flexible stress sensor array. These advantages make the SLMC technology well suited for a wide range of applications, including wearable devices, biomedical sensing, structural health monitoring, and robotics, where accurate and real-time measurement of spatial stress distribution is crucial for performance optimization and safety enhancement. However, it is important to acknowledge that challenges may arise in certain aspects of the technology.

The distributed fiber sensing has the property of brittle fracture. However, distributed fiber sensing provides continuous, real-time monitoring along the entire fiber length, enabling the measurement of strain, temperature, and other physical parameters at high spatial resolution. This makes the SLMC suitable for applications that require distributed sensing over large areas or long distances. Additionally, future advances in optical fiber materials such as the development of bendable fiber will greatly improve the durability and flexibility of the fiber. RLC resonant sensing often requires complicated circuit design, due to resistors, inductors, and capacitors. Future research can combine RLC resonant sensing with other sensing materials to achieve simultaneous measurement of multiple parameters (humidity, temperature, etc.), and simplify circuit design to achieve higher integration. Transmission line sensing still presents challenges in terms of the complexity of fabrication and cost, and future research could simplify fabrication processes by applying screen printing, gravure printing or spin coating and use less expensive materials. Ionic conductor sensing and triboelectric sensing are easily disturbed by environmental factors, and future research can develop new ionic/triboelectric conductor materials to expand its application field and improve its stability.

This review introduces the mechanisms of flexible stress sensors, as well as discusses the advantages and disadvantages of the design of flexible stress sensor arrays. To address the issue of excessive signal wires in current flexible stress sensor arrays, solutions for flexible stress sensor arrays are summarized, and the implementation principle and pros and cons of existing SLMC measurement are analyzed. In summary, flexible stress sensor arrays allow for quantification and analysis of the magnitude and distribution of spatial stress, while also offering advantages such as flexibility, integration, and systematization. However, increasing the number of signal wires may negatively impact the stability and reliability of the sensor array. Thus, it is essential to develop more efficient, reliable, and cost-effective methods for arraying flexible stress sensors to promote their application in various fields.

## Figures and Tables

**Figure 1 micromachines-14-01554-f001:**
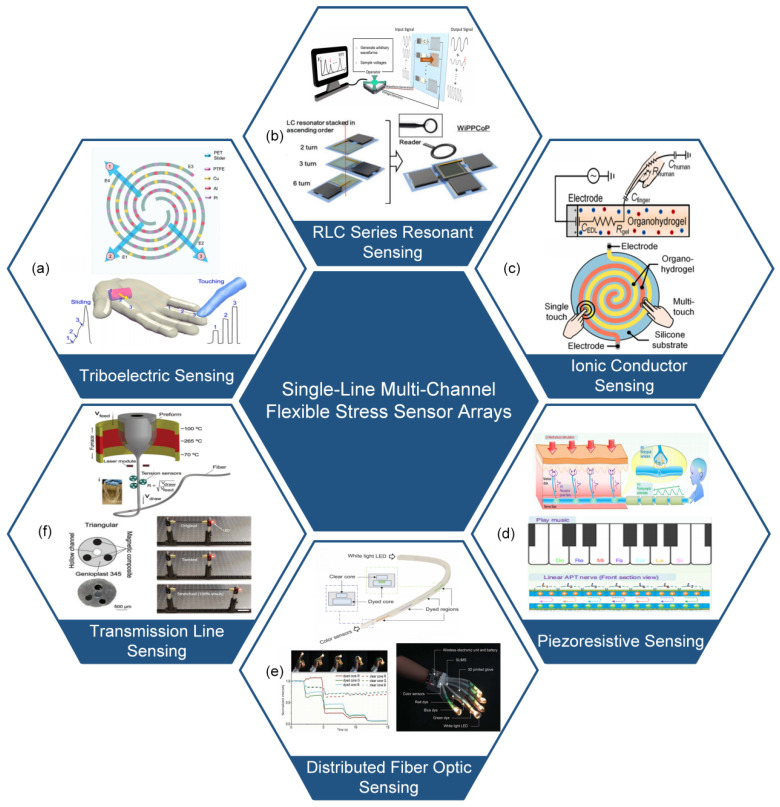
Implementation methods of single-line multi-channel signal measurement. (**a**) Triboelectric sensing (reprinted with permission from [42]; copyright 2022, Elsevier). (**b**) RLC series resonant sensing (reprinted with permission from [28]; copyright 2019, Wiley-VCH). (**c**) Ionic conductor sensing (reprinted with permission from [36]; copyright 2021, Elsevier). (**d**) Piezoresistive sensing (reprinted with permission from [41]; copyright 2020, Springer Nature). (**e**) Distributed fiber optic sensing (reprinted with permission from [34]; copyright 2020, AAAS). (**f**) Transmission line sensing (reprinted with permission from [47]; copyright 2023, Wiley-VCH (reprinted with permission from [48]; copyright 2022, AAAS).

**Figure 5 micromachines-14-01554-f005:**
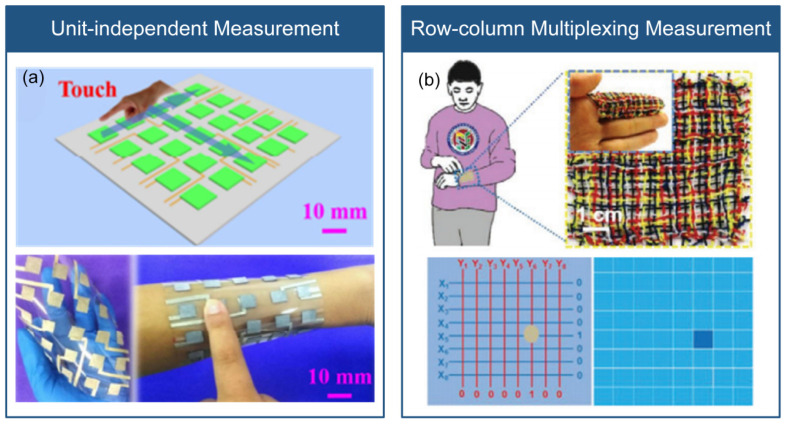
Triboelectric flexible stress sensor array measurement method. (**a**) Unit-independent measurement (reprinted with permission from [70]; copyright 2020, Elsevier). (**b**) Row–column multiplexed measurement (reprinted with permission from [70]; copyright 2020, Wiley-VCH).

**Figure 6 micromachines-14-01554-f006:**
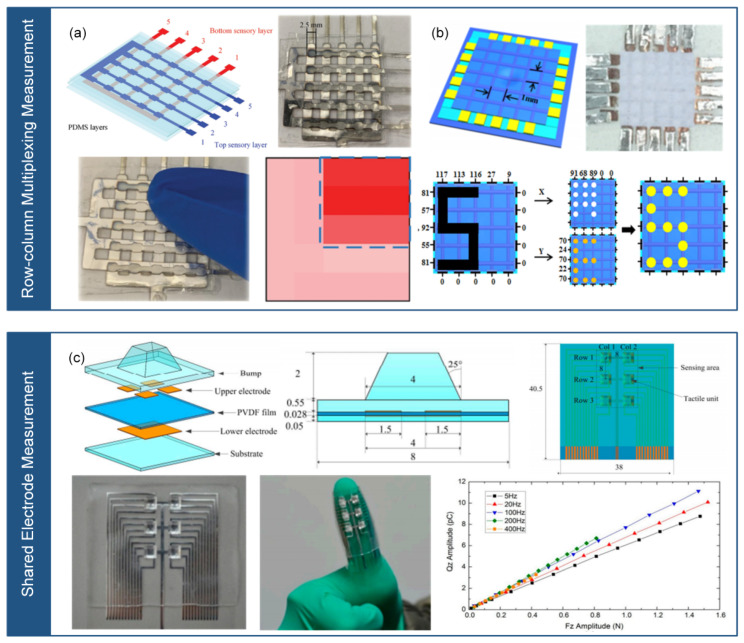
Arrayed measurement methods for piezoelectric flexible stress sensors. (**a**) Row-column multiplexing measurement (reprinted with permission from [85]; copyright 2021, Wiley-VCH). (**b**) Row-column multiplexing measurement (reprinted with permission from [86]; copyright 2021, American Chemical Society). (**c**) Shared electrode measurement (reprinted with permission from [88]; copyright 2016, MDPI). Note: In (**a**,**b**), the same Row-column multiplexing measurement is presented but from different references.

**Table 1 micromachines-14-01554-t001:** The advantages and disadvantages of the existing sensor array measurement method.

Method	Advantages	Limitations
RLC Resonant Sensing	Two signal lines Tunable sensitivity Multi-model sensing	Complicated circuit design Crosstalk between frequencies
Transmission Line Sensing	One signal line High integration	High cost Complicated fabrication Complicated signal decoding
Ionic Conductor Sensing	Two signal lines Multi model measurement	Low temperature stability Complicated signal decoding
Triboelectric Sensing	Two signal lines Low energy consumption High dynamic performance	Poor static measurement Low signal stability
Piezoresistive Sensing	One signal line Simple signal acquisition Low cost	Stress magnitude unmeasurably Multiple-channel unmeasurably
Distributed Fiber Optic Sensing	One signal line High spatial resolution Multi-model detection	Fiber brittleness Complicated signal decoding
Unit-Independent Measurement	High precision	Tedious signal line Complicated design
Row–Column Multiplexed Measurement	High precision Reduced signal line	Signal crosstalk Tedious signal line
Anisotropic Electrical Impedance Tomography	Reduce design complexity	Signal crosstalk Low resolution Complicated signal processing Tedious signal line
Shared Electrode Measurement	Reduced signal line	Complicated design Signal crosstalk Tedious signal line

## Data Availability

Data available on request from the authors.

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
