# Peer review of "Single-Line Multi-Channel Flexible Stress Sensor Arrays"

_micromachines, 2023, doi:10.3390/mi14081554_

Round 1

Reviewer 1 Report

The review has systematically summarized research work on flexible stress sensor arrays, which is of much interest. Especially, the single-wire multi-channel signal measurement has been introduced to settle the excessive signal wires of traditional sensing strategy. The following issues need to be resolved.

1. Authors summarized the work of many articles, but the works of Professor Zhenan Bao and Professor John A. Rogers, who are very active in the field of flexible electronics in recent years, was not mentioned.

2. Authors introduced various single-line multi-channel measurement technologies in the third part, which is scarcely prominent compared to the traditional sensing array in the second part. More attention should be paid to the implementation principles, technical paths, and applications of these technical methods, but the relevant content is very limited.

3. The authors introduced the single-line multi-channel for piezoresistive sensing, but the summary in this section is not very clear and appears irrelevant to the topic.

4. Why is distributed fiber optic sensing introduced at the end of part 3? As is well known, optical fiber has no advantage in the field of flexible sensing due to its brittle fracture characteristics, and optical fiber sensing itself is a sensing technology that cannot be used to solve the problem of excessive signal lines in traditional sensors.

5. The advantages and disadvantages of various technologies should be given.

6. The figures in the paper need to be carefully revised, and some of them are difficult for readers to obtain practical information.

In general, it is fine. further proofreading needs to be practised.

Author Response

Referee 1

The review has systematically summarized research work on flexible stress sensor arrays, which is of much interest. Especially, the single-wire multi-channel signal measurement has been introduced to settle the excessive signal wires of traditional sensing strategy. The following issues need to be resolved.

Response:

We thank the reviewer for spending time evaluating the manuscript and providing constructive comments to improve the manuscript.

1.      Authors summarized the work of many articles, but the works of Professor Zhenan Bao and Professor John A. Rogers, who are very active in the field of flexible electronics in recent years, was not mentioned.

Response:

Thank you very much for the valuable comment. We sincerely appreciate the observation that two highly active experts in the field of flexible electronics in recent years, Professor Zhenan Bao and Professor John A. Rogers, were not mentioned. We apologize for this oversight. We have added citations to Professor Zhenan Bao and Professor John A. Rogers in the introduction section to accurately reflect their important work.

(Paragraph 2, Section 1): The research work of Professor Zhenan Bao and Professor John A. Rogers is of great significance in the field of flexible electronics, particularly in the development of high-performance flexible stress sensors. Their research achievements have provided crucial technical support and guidance for stress sensors in flexible electronic applications. For instance, in their review, professor Rogers et. al[4]. introduced unconventional methods for fabricating and patterning nanomaterials, which laid a new scientific foundation for the fabrication of small-sized and patterned structures. These methods have significant implications in specific domains and in combination with other fabrication techniques. On the other hand, professor Bao's research[5] has made significant progress in the scalable synthesis of multifunctional polyaniline hydrogels and their outstanding electrode performance, offering strong support for the preparation of flexible stress sensors. These studies provide important references for our understanding and application of flexible stress sensors.

2.      Authors introduced various single-line multi-channel measurement technologies in the third part, which is scarcely prominent compared to the traditional sensing array in the second part. More attention should be paid to the implementation principles, technical paths, and applications of these technical methods, but the relevant content is very limited.

Response:

Thank you for the comment and highlighting the concern regarding the presentation of single-line multi-channel measurement technologies in the manuscript. In response to the valuable comment, we have made revisions to ensure a more comprehensive and thorough discussion of these aspects.

(Paragraph 1, Section 3.1): The RLC resonant sensing utilizes the interaction between inductance and capacitance. By adjusting the resonant frequency of the RLC resonant circuit, it is possible to obtain multiple sensor signals using only two signal lines. Implementing this technology requires selecting appropriate inductance and capacitance components, as well as designing and fabricating the circuit.

(Paragraph 3, Section 3.1): RLC resonant sensing has wide applications in areas such as wireless communication, biomedical, and environmental monitoring. In wireless sensor networks, it enables wireless monitoring and data transmission of various physical parameters (temperature, humidity, etc.). In bio-sensors, it can be used to monitor physiological parameters and disease indicators within the human body.

(Paragraph 1, Section 3.2): The transmission line sensing measures the voltage or current on the transmission line to obtain information about the target physical quantity. When the target physical quantity changes, it affects the electromagnetic field distribution on the transmission line, thereby altering the characteristics of the transmission line, such as resistance, inductance, and capacitance. Implementing this technology requires selecting appropriate transmission line types and parameters, as well as designing and manufacturing the sensors.

(Paragraph 1, Section 3.3): The ionic conductor sensing enables the determination of stress magnitude and location information through two single signal lines, providing a low-cost, easily operated for measuring stress distribution in electronic skins, wearable electronic devices. In addition, implementing this technology requires the selection of suitable ion conductor materials and the design of corresponding circuits.

(Paragraph 3, Section 3.3): The ionic conductor sensing finds application potentials in fields such as human-machine interfaces and biosensing.

(Paragraph 1, Section 3.4): Triboelectric sensing is a sensing array composed of triboelectric materials with unique physical and chemical properties (electron affinity, mobility, adhesion energy, and surface potential), which measures the change of the applied stress through the charge variation generated by the friction between the triboelectric materials.

(Paragraph 3, Section 3.4): Triboelectric sensing has the application potential in the fields of friction and wear monitoring and tactile sensing.

(Paragraph 3, Section 3.5): Piezoresistive sensing technology is widely used in the field of structural monitoring. The sensors can be used to monitor stress changes in multiple key locations such as buildings or bridges to assess the health and safety of the structure. In the medical field, piezoresistive sensing technology can be applied to the measurement of stress distribution on human body, such as on mattresses, to improve treatment.

3.      The authors introduced the single-line multi-channel for piezoresistive sensing, but the summary in this section is not very clear and appears irrelevant to the topic.

Response:

Thank you for the comment on the piezoresistive sensing single-wire multichannel summary. We apologize for any confusion caused by the unclear and irrelevant descriptions in this section. We have summarize this in detail in this section to ensure that it provides a comprehensive and relevant overview of the single-wire multi-channel approach to piezoresistive sensing.

(Paragraph 3, Section 3.5): By manipulating the resistance at different positions within the piezoresistive material, piezoresistive sensing-based sensor array can determine the stress location by analyzing the resulting resistance. Piezoresistive sensing has several advantages, such as high sensitivity, a wide measurement range, low cost, ease of manufacturing, and simple signal acquisition circuits. However, for some application scenarios, it is necessary to consider its stress magnitude and multiple-channel unmeasurably. Piezoresistive sensing technology is widely used in the field of structural monitoring. The sensors can be used to monitor stress changes in multiple key locations such as buildings or bridges to assess the health and safety of the structure. In the medical field, piezoresistive sensing technology can be applied to the measurement of stress distribution on human body, such as on mattresses, to improve treatment.

4.      Why is distributed fiber optic sensing introduced at the end of part 3? As is well known, optical fiber has no advantage in the field of flexible sensing due to its brittle fracture characteristics, and optical fiber sensing itself is a sensing technology that cannot be used to solve the problem of excessive signal lines in traditional sensors.

Response:

Thank you for the comment on the brittle fracture characteristics of optical fibers. We agree with the comment. It is worth noting that this work focuses on the review of single-wire multi-channel measurement methods. Although the formation of the present optical fiber may not meet the requirements of flexibility, but the mechanism of this sensor is a practical method to implement the single-wire multi-channel measurement method. And with the development of bendable optical fibre, the durability and flexibility of fibers could be greatly improved. Therefore, advances make it possible to integrate fiber optic sensors into flexible structures and systems, expanding their applicability in the field of flexible sensing.

(Paragraph 6, Section 4): The distributed fiber sensing has the property of brittle fracture. However, distributed fiber sensing provides continuous, real-time monitoring along the entire fiber length, enabling the measurement of strain, temperature, and other physical parameters at high spatial resolution. This makes the SLMC suitable for applications that require distributed sensing over large areas or long distances. And future advances in optical fiber materials such as the development of bendable fiber will greatly improve the durability and flexibility of the fibre.

5.      The advantages and disadvantages of various technologies should be given.

Response:

Thank you for the comment and emphasize the importance of discussing the pros and cons of various technologies. We have added a table highlighting the advantages and disadvantages of each technique discussed.

Table 1. The advantages and disadvantages of the existing sensor array measurement method

Method

Advantages

Limitations

RLC Resonant Sensing

l   Two signal lines

l   Tunable sensitivity

l   Multi-model sensing

l   Complicated circuit design

l   Crosstalk between frequencies

Transmission Line Sensing

l   One signal line

l   High integration

l   High cost

l   Complicated fabrication

l   Complicated signal decoding

Ionic Conductor Sensing

l   Two signal lines

l   Multi model measurement

l   Low temperature stability

l   Complicated signal decoding

Triboelectric Sensing

l   Two signal lines

l   Low energy consumption

l   High dynamic performance

l   Poor static measurement

l   Low signal stability

Piezoresistive Sensing

l   One signal line

l   Simple signal acquisition

l   Low cost

l   Stress magnitude unmeasurably

l   Multiple-channel unmeasurably

Distributed Fiber Optic Sensing

l   One signal line

l   High spatial resolution

l   Multi-model detection

l   Fiber brittleness

l   Complicated signal decoding

Unit-independent Measurement

l   High precision

l   Tedious signal line

l   Complicated design

Row-column Multiplexed Measurement

l   High precision

l   Reduced signal line

l   Signal crosstalk

l   Tedious signal line

Anisotropic Electrical Impedance Tomography

l   Reduce design complexity

l   Signal crosstalk

l   Low resolution

l   Complicated signal processing

l   Tedious signal line

Shared Electrode Measurement

l   Reduced signal line

l   Complicated design

l   Signal crosstalk

l   Tedious signal line

(Paragraph 3, Section 3.1): The RLC sensing has the advantages of two signal lines, tunable sensitivity, and multi-model sensing. However, due to the RLC resonant circuit, the sensor array has a complicated circuit design, and has a crosstalk between different RLC resonators.

(Paragraph 3, Section 3.2): The transmission line sensing only has one signal line, which has the advantage of high integration. While it has drawbacks, including high cost, complicated fabrication, and complexity of information decoding.

(Paragraph 3, Section 3.3): The ionic conductor sensing technology has the advantages of two signal lines and multi model measurement. However, it still needs to address challenges such as low temperature stability and complicated signal decoding.

(Paragraph 3, Section 3.4): Triboelectric sensing technology has the advantages of two signal lines, low energy consumption, high dynamic performance, but it still needs to solve the challenges of poor static measurement and low signal stability.

(Paragraph 3, Section 3.5): Piezoresistive sensing has several advantages, such as one signal line, simple structure, low cost, and simple signal acquisition circuit. However, for some application scenarios, it is necessary to consider its stress magnitude and multiple-channel unmeasurably

(Paragraph 3, Section 3.6): Distributed fiber optic sensing provides significant advantages, including one signal line, high spatial resolution, and multi-model detection. However, it also faces challenges such as fiber brittleness, signal parsing and data processing complexity.

6.      The figures in the paper need to be carefully revised, and some of them are difficult for readers to obtain practical information.

Response:

Thank you for the comment. We have made improvements to Figures 1, 7, 8, and 10 to enhance the readability of the visual information presented in the paper.

Figure 1. Implementation methods of Single wire multi-channel signal measurement. (a) Triboelectric Sensing.(Reprinted with permission from Ref.[23]. Copyright 2022, Elsevier.) (b) RLC Series Resonant Sensing. (Reprinted with permission from Ref.[9] Copyright 2019, Wiley-VCH.) (c) Ionic Conductor Sensing. (Reprinted with permission from Ref.[17] Copyright 2021, Elsevie.) (d) Piezoresistive Sensing. (Reprinted with permission from Ref.[22] Copyright 2020, Springer Nature.) (e) Distributed Fiber Optic Sensing. (Reprinted with permission from Ref.[15] Copyright 2020, AAAS.)

Figure 7. RLC Series Resonant Sensing. (a) Wireless passive LC stress sensor. (Reprinted with permission from Ref.[5] Copyright 2021, Wiley-VCH.) (b) Stress sensor system based on wireless system to differentiate parallel signals. (Reprinted with permission from Ref.[9] Copyright 2019, Wiley-VCH.) (c) Flexible sensor array based on multiple band-pass filters. (Reprinted with permission from Ref.[13] Copyright 2022, Springer Nature.)

Figure 8. Ionic Conductor Sensing. (a) Self-healing composite hydrogel sensors. (Reprinted with permission from Ref.[14] Copyright 2020, Wiley-VCH.) (b) Parallel interdigitated electrode structure-based hydrogel sensors. (Reprinted with permission from Ref.[17] Copyright 2021, Elsevier.)

Figure 10. Piezoresistive Sensing and Distributed Fiber Optic Sensing. (a) Biomimetic sensory nerve sensor. (Reprinted with permission from Ref.[22] Copyright 2020, Springer Nature.) (b) Stretchable distributed fiber optic sensor. (Reprinted with permission from Ref.[15] Copyright 2020, AAAS.)

References:

[1]   Y. Xia, J. A. Rogers, K. E. Paul, and G. M. Whitesides, Unconventional Methods for Fabricating and Patterning Nanostructures, Chem. Rev., 1999, 99, 1823-1848.

[2]   L. Pan, G. Yu, D. Zhai, H. R. Lee, W. Zhao, N. Liu, H. Wang, B. C. Tee, Y. Shi, Y. Cui, and Z. Bao, Hierarchical nanostructured conducting polymer hydrogel with high electrochemical activity, Proc. Natl. Acad. Sci. U. S. A., 2012, 109, 9287-92.

[3]   L. Q. Li, R. J. He, M. S. Soares, S. Savovic, X. H. Hu, C. Marques, R. Min, and X. L. Li, Embedded FBG-Based Sensor for Joint Movement Monitoring, Ieee Sens J, 2021, 21, 26793-26798.

[4]   R. Min, X. Hu, L. Pereira, M. Simone Soares, L. C. B. Silva, G. Wang, L. Martins, H. Qu, P. Antunes, C. Marques, and X. Li, Polymer optical fiber for monitoring human physiological and body function: A comprehensive review on mechanisms, materials, and applications, Optics & Laser Technology, 2022, 147.

Reviewer 2 Report

The review paper focuses on the challenge of excessive signal wires in the current implementation of flexible stress sensor arrays, which results in reduced deformability, stability, reliability, and increased costs. The primary obstacle lies in the electric amplitude modulation nature of the sensor unit's signal, allowing only one signal per wire. To overcome this challenge, the single wire multi-channel signal (SWMC) measurement has been developed, enabling simultaneous detection of multiple sensor signals through one or two signal wires, which effectively reduces the number of signal wires, thereby enhancing stability, deformability, and reliability. This review paper summaries and introduces the development of the SWMC measurement according to their mechanism. In conclusion, a minor revision is suggested before publishing this article.

1. The authors should make a table to make a comparison of the proposed SWMC measurement technology with other existing technologies to highlight its advantages and limitations.

2. The authors should discuss the potential challenges and limitations of the proposed technology and provide recommendations for future research to address these issues.

3. Is the proposed SWMC measurement technology applicable to a wide range of scenarios or is it limited to specific conditions or samples? If so, have the authors discussed the implications of these limitations?

4. The authors should provide more details about the materials and testing equipment used in the study. This would help readers understand the reliability and validity of the proposed SWMC measurement technology.

none

Reviewer 3 Report

This paper reports an interesting review about single-line Multi-Channel Flexible Stress Sensor Arrays. some comments.

1. Stress sensor is not well presented. What is a stress sensor? is it related with stress hormone? Define very well stress in this case.

2. I miss more motivation for this work compared with state of the art.

3. Section 3.6 is weak. Why only distributed optical fiber sensor? where is the quasi-distributed ones? Please improve this section with quasi-distributed like FBGs: https://ieeexplore.ieee.org/abstract/document/9578992

4. I miss more info about technology of optical fiber like intensity-based sensor. Please consider to add some literature related: Optics & Laser Technology 147, 107626, 2022.

5. I miss some future perspectives.

NA

Round 2

Reviewer 3 Report

The paper was well improved and can be published as it is

Author Response

We thank the reviewer for taking the time to evaluate the manuscript and for this positive comment on our work.